# Exploring Early, Middle, and Late Loss in Basic Activities of Daily Living among Nursing Home Residents: A Multicenter Observational Study

**DOI:** 10.3390/healthcare12080810

**Published:** 2024-04-09

**Authors:** Pau Moreno-Martin, Eduard Minobes-Molina, Aina Carbó-Cardeña, Montse Masó-Aguado, Montserrat Solé-Casals, Meritxell Torrents-Solé, Judit Bort-Roig, Jordi Amblàs-Novellas, Xavier Gómez-Batiste, Javier Jerez-Roig

**Affiliations:** 1Research Group on Methodology, Methods, Models and Outcomes of Health and Social Sciences (M_3_O), Faculty of Health Sciences and Welfare, Centre for Health and Social Care Research (CESS), University of Vic—Central University of Catalonia (UVic-UCC), 08500 Vic, Spain; pau.moreno@uvic.cat (P.M.-M.); aina.carbo@uvic.cat (A.C.-C.); montse.maso@uvic.cat (M.M.-A.); javier.jerez@uvic.cat (J.J.-R.); 2Institute for Research and Innovation in Life and Health Sciences in Central Catalonia (IRIS-CC), 08500 Vic, Spain; judit.bort@uvic.cat; 3Spanish Society of Geriatrics and Gerontology, C. Príncipe de Vergara, 57-59, 28006 Madrid, Spain; 4Central Catalonia Chronicity Research Group (C3RG), Faculty of Health Sciences and Welfare, Centre for Health and Social Care Research (CESS), University of Vic—Central University of Catalonia (UVic-UCC), 08500 Vic, Spain; montse.sole@uvic.cat (M.S.-C.); jordi.amblas@uvic.cat (J.A.-N.); xavier.gomez@uvic.cat (X.G.-B.); 5Independent Researcher, 08500 Vic, Spain; meritxell.torrents@uvic.cat; 6Sport and Physical Activity Research Group, Faculty of Health Sciences and Welfare, University of Vic—Central University of Catalonia (UVic-UCC), 08500 Vic, Spain; 7Faculty of Medicine and Chair in Palliative Care, University of Vic—Central University of Catalonia (UVic-UCC), 08500 Vic, Spain; 8Institute of Sport Science and Innovations, Lithuanian Sports University, 44221 Kaunas, Lithuania

**Keywords:** activities of daily living, disability, geriatric syndrome, nursing homes, ageing

## Abstract

Nursing home (NH) residents commonly face limitations in basic activities of daily living (BADLs), following a hierarchical decline. Understanding this hierarchy is crucial for personalized care. This study explores factors associated with early, middle, and late loss in BADLs among NH residents. A multicenter cross-sectional study was conducted in 30 NHs in Catalonia, Spain. Dependent variables were related to limitations in BADLs: early loss (self-care-related BADLs: personal hygiene, dressing, or bathing), middle loss (mobility-related BADLs: walking or wheelchair handling, toileting, and transferring), and late loss (eating). Independent variables were based on a comprehensive geriatric assessment and institutional factors. Logistic regression was used for the multivariate analyses. The study included 671 older adults. Early loss in BADLs was significantly associated with urinary incontinence, cognitive impairment, and falls. Middle loss in BADLs was linked to fecal incontinence, urinary incontinence, ulcers, and cognitive impairment. Late loss in BADLs was associated with fecal incontinence, the NH not owning a kitchen, neurological disease, cognitive impairment, dysphagia, polypharmacy, and weight loss. These findings highlight the need to address geriatric syndromes, especially cognitive impairment and bladder/bowel incontinence. Monitoring these syndromes could effectively anticipate care dependency. The presence of kitchens in NHs may help to address limitations to eating, allowing for potential personalized meal adaptation.

## 1. Introduction

The global population is aging due to increased life expectancy, leading to an unprecedented growth of older adults, a significant portion of whom will require nursing home (NH) services [1,2]. Although community-based care delivery systems aim to enable older adults to remain at home, NH services are under growing pressure to accommodate an increasingly frail population, which is placing significant strain on available resources [3].

When an individual’s care needs surpass what home settings can offer, it frequently prompts a transition to an NH for more specialized support [3]. Indeed, the primary predictors of NH admission are the loss of independence in more than three basic activities of daily living (BADLs) [4]. Activities of daily living (ADLs) can be divided into basic self-care skills such as eating, bathing, or dressing (BADLs), more complex and instrumental activities such as using a telephone, doing the laundry, or managing medications (IADLs); and advanced cultural and gender-specific activities not necessary for independent living such as hobbies, religion, and working (AADLs) [5].

Most NH residents experience diverse degrees of limitations or dependence in performing BADLs, and this reliance tends to increase over time [6]. High rates of increase in dependence, spanning from 38.9% to 50.6%, have been reported within a one-year timeframe [6]. This growing dependency among NH residents is associated with lower quality of life [7], recurring hospitalizations [8], increased healthcare usage [9], and higher all-cause mortality [10].

Activities of daily living exhibit a hierarchical pattern of decline, where AADLs typically wane before IADLs, and IADLs precede BADLs [11]. In the context of BADLs, scientific studies that investigate BADLs’ decline consistently underscore a trend: bathing commonly diminishes earliest, while feeding tends to persist the longest. However, consensus on the sequencing of other BADLs remains elusive [12]. Five decades ago, Katz and colleagues (1963) pioneered the sequence of BADL loss in community-dwelling older adults, establishing the order as bathing, dressing, toileting, transferring, continence, and eating [13]. This sequence was later validated in a recent NH study [14]. The hierarchy proposed by Katz is now used to categorize BADLs into three groups: “early loss” (self-care BADLs), including bathing, dressing, and personal hygiene; “middle loss”, involving mobility-related BADLs like toilet use, transfer, and mobility; and “late loss”, specifically related to eating [15,16,17].

The hierarchy of loss of BADLs is argued to be driven by its complexity, which encompasses physical demands, decision making, survival, and cultural roles. Physically demanding BADLs requiring greater strength, dexterity, and balance tend to decline first [18]. For example, bathing, with its multiple subtasks involving grasping fixtures, bending, and drying off, requires coordination of balance, strength, and dexterity, while eating primarily relies on minimal manual dexterity. In addition, the hierarchy follows a pattern transitioning from BADLs involving both upper and lower limb engagement (e.g., bathing and dressing), to those primarily reliant on lower limb functions (e.g., walking and toileting), and finally to activities predominantly using the upper limbs (e.g., eating) [18]. Furthermore, higher-level decision making plays a role, with BADLs involving more complex decisions tending to be lost earlier [19]. For example, bathing entails the need for planning, involving steps such as wetting, using soap, washing various body areas, and coordinating these actions, while transferring primarily involves initiating the task of moving from the bed to an armchair. Lastly, BADLs crucial for survival, such as walking and eating, are typically retained longer than those influenced by cultural factors, like personal hygiene, dressing, and bathing [19].

Studies conducted in NHs commonly focus on overall BADL scores and give minimal consideration to individual BADLs [17,20]. This approach can overlook variations in disability levels and provide a less comprehensive evaluation of specific factors [17,20]. A deeper understanding of the hierarchical process of loss of BADLs can serve as a roadmap for more effective and personalized care. It may assist in the accurate prediction and identification of care requirements, shed light on the underlying mechanisms of BADL impairment, and provide valuable guidance for interventions [14,20,21,22]. Consequently, this understanding can significantly reduce the caregiving burden and improve patient safety and overall quality of life [14].

To our knowledge, no study in NHs has examined the factors associated with each BADL category: early loss, middle loss, and late loss. Each category entails distinct physical demands and decision-making requirements and holds different cultural and survival roles. Thus, investigating each BADL level individually is crucial to gain a more comprehensive understanding of their unique intricacies and challenges. Consequently, this study aims to analyze the factors associated with limitations in each BADL category among NH residents.

## 2. Materials and Methods

### 2.1. Study Design

A multicenter cross-sectional study, encompassing the 10 health regions in Catalonia, Spain, was conducted across 30 NHs. The data collection period extended from December 2021 to June 2022, occurring approximately one and a half years after the World Health Organization declared the COVID-19 pandemic on 11 March 2020 [23]. This study is a subset of the larger project called Resicovid [24], and its design adheres to the STROBE (Strengthening the Reporting of Observational Studies in Epidemiology) guidelines for cross-sectional studies [25]. Ethical permission was obtained by the Ethics and Research Committee of the University of Vic—Central University of Catalonia (registration number 181/2021 approved on 5 October 2021).

### 2.2. Sample

To ensure a representative sample, NHs in Catalonia were stratified by size (small, medium, large), ownership (for-profit, non-profit, partial for-profit/non-profit), and health region representation (outlining the organization of health services in Catalonia, which categorizes its territory according to geographical, socioeconomic, and demographic factors). Simple random sampling was used within each stratum, resulting in the selection of 30 NHs out of a total of 779 in Catalonia. Residents from medium and large NHs were subject to random sampling, with 25% of residents selected for participation. In small NHs (those with a capacity of ≤50 residents), all eligible residents were recruited. There were no additional eligibility criteria beyond the willingness to participate and being selected in the sampling process. In the random sampling, a waiting list of residents was created. If a selected resident declined participation, the next resident randomized on the waiting list was included in the sample. 

### 2.3. Study Procedures

Prior to data collection, selected NHs were contacted via email and telephone. A project overview video was shared, explaining its purpose, implications, and participation prospects. Subsequent online meetings were conducted to address queries, coordinate data collection logistics, and confirm scheduled dates. In instances where a NH declined participation, the subsequent facility on the randomized list was contacted. Participants or their legal guardians were informed about the project, and consent was obtained from those willing to participate. In-person data collection proceeded as scheduled and was conducted by two qualified project personnel, either physiotherapists or nurses, employing tablet-based online questionnaires. To adhere to COVID-19 containment measures, resident data collection was conducted through proxies. Professionals chosen to complete questionnaires were selected based on their specific expertise related to the variables in question. This selection process was pre-established and organized during the previously mentioned meeting. Institution-related variables were consistently sourced from the NH directors.

### 2.4. Data Collection

The evaluation of performance of BADLs was conducted with the modified Barthel index [26]. Urinary and fecal incontinence items were omitted from the evaluation, given their classification as geriatric symptoms rather than integral components of BADLs [6]. The assessment included rating the following items on a Likert-type 5-point scale: eating, personal hygiene, dressing, bathing, transferring, walking or wheelchair handling, toileting, and climbing stairs. Three dependent variables were derived: early loss (encompassing self-care-related BADLs), middle loss (related to mobility-related BADLs), and late loss (pertaining to eating). These variables were categorized based on their presence or the absence of limitations. Both ‘minimal help required’ (score of 4) and ‘totally independent’ (score of 5) were categorized as not limited [27]. Limitation in self-care (early loss) was considered when any of the following BADLs had a score below 4: personal hygiene, dressing, or bathing. Limitations in mobility was indicated when any of the following BADLs had a rating below 4: walking or wheelchair handling, toileting, or transferring. Limitation in eating was inferred with ratings below 4 in the eating BADL.

Furthermore, the data collection encompassed various sociodemographic variables, including age, sex, duration of institutionalization (months), educational level (categories REF), and marital status (categories). Additionally, a comprehensive geriatric assessment was conducted using the Frail-VIG index [28], encompassing a broad spectrum of factors. This assessment specifically explored IADLs, evaluating the ability to manage money (requiring assistance managing financial matters, e.g., banks, shops, restaurants), the phone (requiring aid in telephone usage), and handling medication (requiring support in medication preparation or administration)—tasks chosen for their universality and lack of cultural or gender biases. Nutritional status was evaluated based on a ≥5% weight loss or reports of decreased appetite over the previous six months [29]. Emotional indicators encompassed the identification of depressive syndrome (feelings of sadness and apathy within the last month) and the presence of insomnia/anxiety (nervousness and sleep difficulties experienced in the last month). The social dimension was investigated through an assessment of loneliness and social isolation, indicated positively if the professional considered that the NH resident experienced social isolation and/or loneliness. Symptoms of pain and dyspnea, as well as sensory impairments related to vision and hearing, were evaluated using a Likert scale ranging from 0–10, where 0 denoted the absence of symptoms or sensory deficits and 10 represented the maximum level [28].

Chronic diseases were assessed, encompassing a spectrum of conditions including oncological, respiratory, cardiovascular, neurological, hepatic, digestive, and renal disorders. Lastly, geriatric syndromes were comprehensively considered, incorporating an examination of delirium (presence of delirium and/or behavior disorder requiring antipsychotic drugs), falls (≥2 falls or hospitalization due to a fall), polypharmacy (taking ≥5 medications), ulcers (presence of ulcers), and dysphagia (difficulty swallowing when eating or drinking; presence of aspiration respiratory infections) over the preceding six months. Cognitive impairment was quantified utilizing Reisberg et al.’s Global Deterioration Scale (GDS), with seven levels from 1 (no cognitive decline) to 7 (very severe cognitive decline) [30]. Bladder and bowel continence status during the previous five days were assessed through Section H of Minimum Data Set (MDS) V.3.0, categorizing residents as continent, occasionally incontinent, frequently incontinent, always incontinent, and not classifiable (due to the use of a catheter, ostomy, or other appliances) [31].

Additionally, institutional-type variables were collected. These variables encompassed residents’ access to outdoor spaces within their buildings and whether these areas featured natural elements like gardens, trees, or plants. Data on the numbers of shared rooms and total rooms were gathered to calculate the ratio of shared rooms to the total. Similarly, we recorded the number of shared bathrooms in relation to the total bathrooms to determine the ratio of shared bathrooms. The type of food service for resident meals, whether provided by catering or managed in-house, was noted, as well as whether the residence was for profit or non-profit. Furthermore, data on staffing were collected, including the numbers of nurses, physiotherapists, occupational therapists, psychologists, social workers, geriatric caregivers, aides, monitors, specialist physicians, and general practitioners, along with their weekly hours. To determine the staffing ratio, the total cumulative hours for each role was divided by the overall number of residents.

### 2.5. Data Analysis

Descriptive analysis was undertaken indicating absolute and relative frequencies for categorical variables and mean and standard deviation (SD) for quantitative variables. Bivariate analysis was performed using the Chi-square test for the categorical variables, and the Student *t*-test (or non-parametric Mann–Whitney test) was used for quantitative variables. For bivariate analyses, ordinal variables with more than two categories were dichotomized using their means, with the exception of urinary and fecal incontinence, which were dichotomized into fully continent versus other categories. However, in the multivariate analyses, these variables were retained in their ordinal form. Multivariate analysis was performed using logistic regression, with a significance level set at 0.05. Variables with a *p*-value of less than 0.25 in the bivariate analysis were considered as potential candidates for inclusion in the multiple models. Backward selection was employed to introduce covariates into the model. The variables ‘age’ and ‘gender’ were initially included and retained throughout the backward selection process. Variables with more than 5% of the total sample missing were excluded from the analyses. Multicollinearity was assessed using a variation inflation factor (VIF) threshold set at 5. Risk measurements were reported as odds ratios (ORs) along with their respective confidence intervals (CIs) and *p*-values. The adjustment of the final model was tested with the Hosmer–Lemeshow test. Statistical Package for the Social Sciences (SPSS) version 29 (SPSS Inc., Chicago, IL, USA) was used for all data analysis.

## 3. Results

### 3.1. Sample Characteristics

The sample comprised 671 older adults, primarily female (75.0%), with an average age of 85.9 years and institutionalization duration averaging 36.6 months. Physical restraints were used by 102 (15.2%) residents as part of their care. Additionally, 425 individuals (66.8%), had a history of COVID-19 infection, and during lockdown, 419 participants (66.0%) were confined to their rooms for an average of 31.5 days. Figure 1 shows the flowchart of the sampling process.

In total, 31.9% of the residents were experiencing multiple chronic diseases and 86.7% took more than five medications. Over the previous 6 months, various health issues were reported, including involuntary weight loss (13.7%), appetite loss (8.5%), and pain (27.3%). Cognitive function was normal in 20.3% of the sample, and 37.3% displayed severe decline. Depressive symptoms were reported by 27.4% of the residents, with 19.5% experiencing anxiety. Loneliness affected 19.2%, while 26.4% experienced either social isolation or loneliness.

Regarding functional status, the majority required assistance in IADLs, with 76.9% needing help with medication, 73.9% with phone use, and 84.2% with handling money. In terms of BADLs, out of a maximum score of 80, 14.9% had a Barthel score of 70 or higher, while 27.5% scored 10 or lower. The average Barthel score across participants was 36.3. In terms of specific BADLs, 86.3% were limited when bathing, 81.7% when climbing stairs, 78.4% when dressing, 72.4% with personal hygiene, 64.7% with toileting, 53.1% when walking, 51.3% with transferring, and 25.8% when eating. Regarding BADL categorizations, 89.6% had limitations on self-care (early-loss BADLs), 59.5% on mobility (middle-loss BADLs), and 28.0% on eating (late-loss BADLs).

In the institutional context, a significant proportion of residents (80.2%) occupied private NH accommodation. Almost all residents (97.6%) were situated in facilities offering indoor access to outdoor spaces, and a substantial number (71.9%) enjoyed the presence of garden-like areas. Shared rooms constituted 69.2% of the available accommodation, with shared bathrooms comprising 52.4%. Concerning meal services, the majority of residents (79.3%) were situated in NHs where the facility managed the kitchen, while a smaller percentage (20.7%) relied on catering services. Regarding staffing in NHs, the bulk of care was delivered by caregivers, averaging 11.8 h per week per resident. Registered nurses and nurse assistants followed with an average of 1.35 and 1.34 h, respectively. Physiotherapists, psychologists, occupational therapists, and social workers contributed in the range of 0.2 to 0.6 h, while geriatricians provided an average of 0.2 h per resident per week. Additional descriptive analyses can be found in Table 1.

### 3.2. Factors Associated with Early-Loss BADLs: Self-Care

For early-loss BADLs (self-care), the bivariate analysis, detailed in Table 2, revealed associated factors, including urinary incontinence, cognitive function, falls, fecal continence, outdoor accessibility, dysphagia, ulcers, confusion disorder, NH type, age, and intellectual disability. Subsequently, in the multivariate analysis, with sex and age as control variables, with an acceptable pseudo-R squared value of 28.0%, only three factors retained statistical significance: urinary incontinence (OR: 1.42), cognitive function (OR: 1.67), and falls (OR: 3.42).

### 3.3. Factors Influencing Middle-Loss BADLs: Mobility

For middle-loss BADLs (mobility), as indicated in Table 3, bivariate analyses showed associated factors encompassing fecal incontinence, ulcers, cognitive function, urinary incontinence, physical restraint, severe mental disorder, dysphagia, outdoor accessibility, age, confusion disorder, institutionalization months, constipation, percentage of caregivers per residents, and days of room isolation due to COVID-19. In the multivariate analyses, adjusting for sex and age, with a pseudo-R squared value of 27.4%, the following factors demonstrated statistical significance: fecal incontinence (OR: 1.41), ulcers (OR: 4.01); cognitive function (OR: 1.14), and urinary incontinence (OR: 1.17).

### 3.4. Factors Related to the Late-Loss BADL: Eating

Turning to the late-loss BADL (eating), as highlighted in Table 4, bivariate analysis revealed associations with numerous factors, including fecal incontinence, food service arrangement, neurological disease, cognitive function, dysphagia, polypharmacy, weight and/or appetite loss, physical restraint, confusion disorder, ulcers, urinary incontinence, NH type, percentage of shared toilets, percentage of caregivers per residents, percentage of doctors per residents, emotional symptoms, renal disease, shortness of breath, sex, marital status, and outdoor accessibility. In the subsequent multivariate analysis, controlling for sex and age, with a competent pseudo-R squared value of 42.8%, several factors retained their statistical significance: fecal incontinence (OR: 1.56), food service arrangement (OR: 4.28), neurological disease (OR: 2.11), cognitive function (OR: 1.31), dysphagia (OR: 2.16), and weight and/or appetite loss (OR: 1.90); polypharmacy (OR: 0.49) was the only significant variable inversely associated with late-loss BADLs (eating) (Table 4). Figure 2 illustrates the associated factors in the multivariate analyses for each dependent variable.

## 4. Discussion

This study examined the associated factors related to the categorizations of early loss, middle loss, and late loss of BADLs among NH residents in Catalonia, Spain. Our findings revealed that early-loss BADLs (self-care related) presented associations with urinary incontinence, cognitive impairment, and falls. Middle-loss BADLs (mobility related) were associated with fecal incontinence, ulcers, cognitive impairment, and urinary incontinence. The late-loss BADL (eating) was associated with fecal incontinence, neurological disease, cognitive impairment, dysphagia, polypharmacy, weight and/or appetite loss, and the NH owning a kitchen.

Most of the participants within this sample of NH residents were care-dependent, matching with the high dependency rates reported in the scientific literature for this population [6]. In addition, the patterns of limitations observed across each of the BADLs align with the hierarchical loss pattern described by Katz [14]. BADLs linked to self-care, such as bathing, dressing, and personal hygiene, exhibited the highest limitation rates (ranging from 72.4% to 86.5%), indicative of early loss of BADLs. Subsequently, BADLs related to mobility, like toileting, walking, and transferring, showed intermediate limitation rates (ranging from 51.3% to 64.7%), falling between self-care BADLs and eating in terms of limitations. Finally, eating showed the lowest limitation rate at 28%, indicative of late-stage loss of BADLs.

Remarkably, climbing stairs, despite having the second-highest limitation rate among BADLs, was excluded from both the hierarchy of BADL loss categorization and the analysis. This omission can be attributed to several factors. Climbing stairs is frequently disregarded in studies examining the hierarchy of loss among BADLs, primarily because it is not typically included in BADL assessment instruments commonly used in NH settings [6,14]. Furthermore, as a mobility-related BADL, it does not neatly align with the other mid-loss BADLs. While it involves minimal decision-making, it exerts higher physical demands than self-care BADLs (early-loss BADLs) [18,19,32]. Assessing climbing stairs using reported tools like the Barthel index can be a contentious issue, primarily due to safety measures that restrict many NH residents from partaking in this activity. Additionally, the presence of elevators in NHs may lead residents to opt for not climbing stairs, even if they are physically capable of doing so. These factors have the potential to introduce confusion for reporting professionals and may impact the accuracy of stair-climbing assessments. In fact, it has been suggested to either eliminate or revise this item to enhance the psychometric performance of responses to the Barthel index items [32].

The factors linked to limitations in each BADL categorization were predominantly geriatric syndromes, encompassing cognitive impairment, urinary and fecal continence, falls, pressure ulcers, dysphagia, polypharmacy, and weight loss. Remarkably, only two factors, namely food service arrangement and neurological disease, departed from this trend and were linked to limitations with eating. This confirms findings from Lane et al. (2017), which demonstrated that geriatric syndromes had stronger links to disability in NH residents compared with chronic conditions [33]. When geriatric syndromes were excluded from their analysis, it significantly reduced the model’s capacity to explain variations in residents’ disability, ranging from 62.7% to 11.2%, indicating that geriatric syndromes contributed to approximately 50% of the unique variation in resident disability. Furthermore, the geriatric syndromes they identified as the most influential factors linked to disability closely paralleled the findings of the current study: balance impairment, cognitive impairment, urinary and bowel incontinence, and pressure ulcers [33]. A possible mechanism for this finding is that geriatric syndromes mediate some of the effects of chronic conditions on disability [33].

Cognitive impairment stands out as the sole factor consistently achieving statistical significance in all three BADL categorizations, similar to previous literature reviews [6]. A meta-analysis underscored a distinct correlation between cognitive impairment and challenges in managing BADLs, proposing that functional decline, like cognitive decline, exists on a continuum from healthy aging to dementia onset [34]. Furthermore, a systematic review identified cognitive impairment as the predominant risk factor for functional decline among NH residents [6].

The statistical significance of incontinence, both urinary and fecal, is evident across all three BADL categorizations. Specifically, early loss of BADLs correlates with urinary incontinence, mid loss with both urinary and fecal incontinence, and late loss with fecal incontinence. This might indicate that incontinence follows the same continuum as cognitive and functional decline, where urinary continence tends to diminish earlier, preceding the occurrence of fecal incontinence. This observation concurs with the established scientific literature, highlighting that fecal incontinence often coexists with an advanced state of frailty and urinary incontinence, while isolated urinary incontinence is more prevalent [35,36,37]. Moreover, the literature consistently highlights the correlation between incontinence and BADL impairments. In fact, there is a specific type of incontinence known as “functional incontinence”, where the root issue lies in the inability to access an appropriate place to excrete (mid-loss BADL) or the dexterity to undress, dress, and clean effectively (early-loss BADLs), rather than a physiological inability to retain urine or feces [36]. Incontinence is frequently assessed within BADL instruments, but given its status as a geriatric syndrome, it is necessary to consider it independently of BADLs [6].

The association between limitations in BADLs and incontinence may operate bidirectionally, acting as a mutual risk factor. Incontinence can contribute to the onset of BADL impairment, and conversely, BADL impairments may serve as a risk factor for the development of incontinence [6,37,38,39]. Coll-Planas et al. (2008) not only proposed incontinence and disability as risk factors for each other but also suggested a shared set of risk factors, such as white matter changes, stroke, and other neurological conditions [39]. Additionally, they presented a comprehensive conceptual framework for the multifactorial etiology of geriatric syndromes. This framework contends that conditions like incontinence and disability arise when impairments span multiple domains, including the lower and upper extremities, the sensory system, and mood. These collective impairments compromise the individual’s ability to compensate for or cope with these challenges, ultimately leading to the manifestation of conditions like incontinence and disability [39]. This framework may further substantiate the notion that geriatric syndromes play a mediating role in the impact of chronic conditions on disability [33].

Early loss of BADLs, those related to self-care, was correlated with a recent history of 2 or more falls, or a fall requiring hospitalization within the last 6 months, along with urinary incontinence and cognitive impairment. This association aligns with existing published research, where history of falls as a risk factor for decline in BADLs has been identified [6]. Specifically, a longitudinal study evaluating risk factors for the decline in personal hygiene identified bowel and bladder incontinence, balance dysfunction, and falls within a 2- to 6-month timeframe, closely aligning with the findings of the current study [40]. This association could be explained by safety-focused care from staff and institutional policies, leading to practices such as activity restrictions or restraint use to minimize short-term falls [41]. Unfortunately, these measures not only directly amplify care dependency but also negatively impact quality of life, contribute to deconditioning, and ultimately elevate the long-term risk of falling [42]. Self-care BADLs, being the most complex, may be promptly restricted in anticipation of a potential future fall.

Limitations in mid-loss BADLs, those related to mobility, were associated with presenting a pressure ulcer in the last 6 months in conjunction with cognitive impairment and urinary and fecal incontinence. This correlation has been consistently observed in published research, as impaired mobility is widely recognized as the foremost host risk factor predisposing individuals to ulcer development [43,44]. Pressure ulcers are commonly regarded as an indicator of the quality of care provided [45,46]. Their occurrence is largely preventable through ensuring proper mobilization and implementing preventive measures [44]. However, healthcare workers often demonstrate suboptimal utilization of effective clinical guidelines due to inadequate education and the high workloads that pose a threat to their effective application [47,48].

Eating, considered the late-loss BADL, correlated with two additional geriatric syndromes—dysphagia and unintentional weight loss—alongside previously discussed issues like fecal incontinence and cognitive impairment. The current scientific literature consistently identifies dementia, unintentional weight loss, and dysphagia as key factors associated with limitations to eating [49,50]. A cross-sectional study found similar associated factors and provided two interpretations that point in distinct directions: one suggesting that those with self-feeding limitations experience inadequate nutritional care leading to malnutrition, and the other proposing that self-feeding limitations in residents may indicate a precursor of approaching the end of life [51].

Notably, beyond the previously mentioned geriatric syndromes, neurological disease and the NH owning a kitchen were identified as additional factors associated with limitations to eating. Parkinson’s disease, excluding Alzheimer’s, stands out as the predominant neurodegenerative disorder, with residents suffering from it exhibiting poor functional status ranging from 68.24% for eating to 99.25% for bathing [52]. Additionally, it has been recognized as a risk factor for future functional decline [6]. The presence of kitchens in NHs has not been addressed in relevant publications regarding dependence or limitation when eating. A cross-sectional study on self-feeding did not identify NH-level explanatory variables, but suggested latent variables within the NH cluster might influence outcomes [51]. Nevertheless, NHs with kitchens have the potential to address individual factors influencing residents’ eating behaviors and food choices, including innate and learned food preferences, motivations, attitudes, hunger, personality traits, and values, among other considerations [50,53].

Polypharmacy, typically expected in frail populations, paradoxically showed a negative association with limitations to eating, suggesting that individuals with higher limitations to eating often have fewer prescribed medications. This observation is supported by a recent scoping review on observational studies about deprescribing in NHs, highlighting that medication discontinuation is strongly influenced by intrapersonal factors, many of which are prevalent in those with limitations to eating [54]. Noteworthy factors specific to NH residents with limitations to eating include practical challenges in medication administration (e.g., in cases of difficulty swallowing or severe dementia), a cautious approach stemming from a higher predisposition to adverse events, and a shorter prognosis, making them less likely to benefit from continued medication treatment [54]. Factors such as sudden weight loss, loss of appetite, and limitations to BADLs are typically considered for prognosis assessment [54].

The practical implications of this study are noteworthy, providing valuable insights to improve care dependency in NH residents across various stages. The identified factors highlight the strategic potential of targeting geriatric syndromes to reduce care dependency. These syndromes might act as mediators, shaping the impact of chronic conditions on care dependency. By potentially compromising individuals’ coping mechanisms in response to health challenges, these syndromes may play a pivotal role in the onset and progression of care dependency [33,39]. Cognitive impairment and bladder/bowel incontinence are critical geriatric syndromes that warrant special attention. They exist on a spectrum from optimal function to total dysfunction, closely aligning with the categories in the hierarchy of loss of BADLs. To preemptively address potential care dependency, it is essential to institute monitoring programs or services tailored to these syndromes [55,56].

Inadequate care provision, noticeable in issues like pressure ulcers, falls, and malnutrition, may be linked to care dependency or limitations to BADLs. Outdated care approaches, centered on survival and safety, jeopardize the well-being of older individuals [57]. Care often prioritizes basic needs, such as bathing or dressing, overshadowing broader wellbeing objectives. A prevalent task-oriented approach, with professionals doing tasks for individuals rather than with them, can lead to a downward spiral—greater loss of functions and increased care consumption [58]. Data from this study affirm this care model, revealing a predominant reliance on caregivers. Registered nurses and nursing assistants operate under restricted hours, while the presence of physiotherapists, psychologists, occupational therapists, social workers, and geriatricians is minimal. Poor quality of NH care has often been associated with insufficient staffing levels, but there is no consistent evidence of a positive relationship between the quantity of staff and quality of care [59].

In light of these challenges, a paradigm shift in long-term care is crucial, emphasizing the optimization of physical and cognitive capacity through a person-centered approach that fosters active involvement and participation of older adults [57,58]. Notably, interventions aimed at improving care involving BADLs have shown promising results [60,61], even reducing the number of falls [42]. Additionally, exercise and physical activity provide clinical benefits across a broad spectrum of both physical and cognitive capacities, effectively addressing various diseases and disabilities in older adults [62,63]. Therefore, it is imperative that the prescription of exercise becomes a mandatory component in the training of all healthcare professionals and is consistently promoted in long-term care facilities [62,63]. Finally, access to a kitchen in NHs may significantly address malnutrition among residents who are dependent when eating, enabling meal adaptation based on factors like innate preferences, motivations, attitudes, hunger, personality traits, and values [50,53].

While the current study provides valuable insights into care dependency among NH residents at different stages, it is essential to recognize its limitations. The cross-sectional design hinders the establishment of cause-and-effect relationships between variables, requiring longitudinal designs for a more in-depth analysis of these pathways. Moreover, reliance on self-reported data from healthcare professionals introduces potential biases, such as recall or reporting bias. Additionally, certain institution-related variables, particularly those related to professional ratios, were excluded from the multivariable analyses due to missing data. Finally, post-COVID-19 data collection involved participants with a history of infection and room confinement, potentially skewing and limiting the study’s results.

Despite these limitations, the study’s strength lies in its comprehensive examination of the factors linked to limitations in NH residents across three categories of BADLs: early loss, middle loss, and late loss. The robust sample size, encompassing a diverse group of NH residents and institutions with minimal limitations in inclusion criteria, adds strength to the findings. Additionally, the analysis of associated factors considers a broad spectrum of variables based on a comprehensive geriatric assessment, including those related to institutional characteristics. The use of the modified Barthel index facilitated the assessment of each BADL, with incontinence, a geriatric syndrome, being considered independently from BADL despite their strong associations [6].

## 5. Conclusions

This study uncovered distinct associations within the different categorization of the hierarchy of BADL loss in NH residents. Early-loss BADLs, related to self-care, demonstrated links to urinary incontinence, cognitive impairment, and falls. Middle-loss BADLs, related to mobility, exhibited associations with fecal and urinary incontinence, ulcers, and cognitive impairment. Late-loss BADLs, specifically capacity to eat, were associated with fecal incontinence, the NH owning a kitchen facility, neurological disease, cognitive impairment, dysphagia, polypharmacy, and weight and/or appetite loss. These insights underscore the importance of targeting geriatric syndromes, particularly cognitive impairment and bladder/bowel incontinence. Monitoring these syndromes might serve as an effective proxy for anticipating care dependency. A shift toward tailored long-term care is essential, with promising interventions focusing on improving care involving BADLs. Moreover, granting kitchen access in NHs might address dependence when eating or malnutrition by facilitating meal adaptation according to individual preferences and values.

## Figures and Tables

**Figure 1 healthcare-12-00810-f001:**
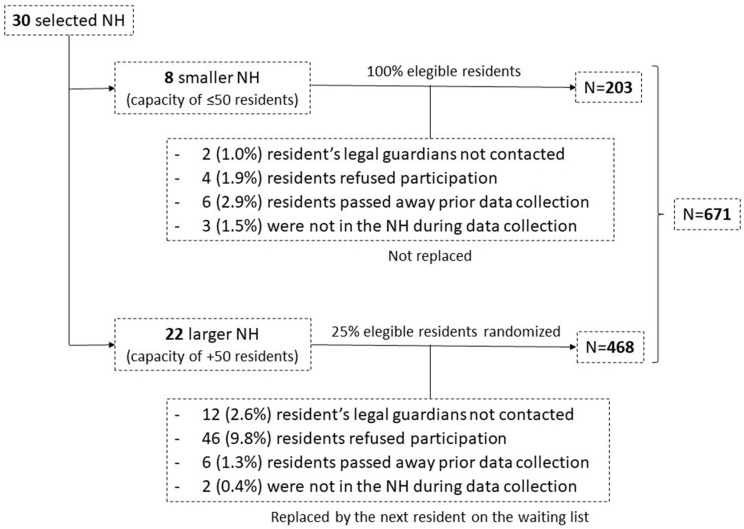
Flowchart illustrating the sampling process for NH residents within the Resicovid project; NH: Nursing home.

**Figure 2 healthcare-12-00810-f002:**
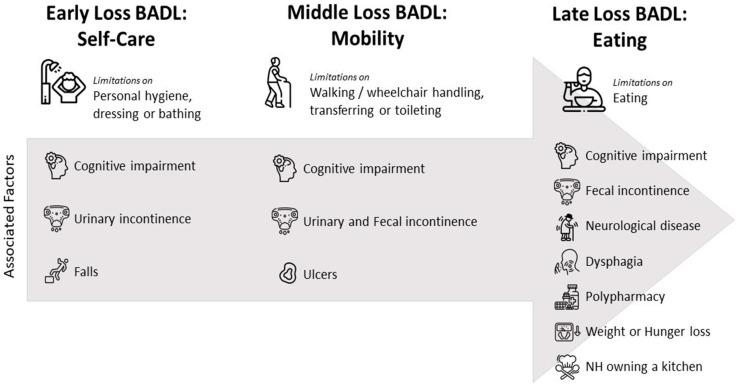
Factors associated with each dependent variable in the multivariate analyses: Early-loss BADLs, mid-loss BADLs, and late-loss BADL; BADLs: basic activities of daily living; NH: nursing home.

**Table 1 healthcare-12-00810-t001:** Demographic, Health, and Institutionalization Characteristics of Older Adults in 30 Nursing Homes in Catalonia, Spain.

Variable (Non-Missing N)	Frequency (%)/Mean (Standard Deviation)
**Sociodemographic variables **	
Sex (671)	
Female	503 (75.0%)
Age (670)	85.87 (±8.75)
Institutionalization months (671)	36.55 (±42.36)
Education level (606)	
Illiterate	105 (15.6%)
Marital Status (671)	
Married/In a relationship	106 (15.8%)
**Health variables **	
Days of room isolation due to COVID-19 (635)	20.77 (±27.74)
Physical restraint (635)	102 (16.1%)
Resident COVID-19 infection history (634)	425 (63.3%)
Cancer (671)	59 (8.8%)
Neurological disease (671)	217 (32.3%)
Digestive disease (671)	40 (6.0%)
Renal disease (671)	142 (21.2%)
Cardiac disease (670)	288 (43.0%)
Severe mental disorder (670)	107 (16.0%)
Intellectual disability (671)	41 (6.1%)
Polypharmacy (670)	581 (86.7%)
Weight loss or appetite loss (671)	100 (14.9%)
Pain symptoms (671)	183 (27.3%)
Shortness of breath (671)	50 (7.5%)
Confusion or behavior disorder (670)	160 (23.9%)
Multiple falls or serious fall (670)	84 (12.5%)
Ulcer or chronic wound (670)	57 (8.5%)
Dysphagia (670)	86 (12.8%)
Urinary continence (669)	
Continent	175 (26.1%)
Usually continent	121 (18.0%)
Occasionally incontinent	17 (2.5%)
Often incontinent	35 (5.2%)
Always incontinent	312 (46.5%)
Not classifiable	2 (0.3%)
Fecal continence (662)	
Continent	351 (52.3%)
Usually continent	79 (11.8%)
Occasionally incontinent	7 (1.0%)
Often incontinent	5 (0.7%)
Always incontinent	223 (33.2%)
Not classifiable	4 (0.6%)
Constipation (668)	263 (39.2%)
Vision impairment (668)	85 (12.7%)
Hearing impairment (667)	119 (17.8%)
Anxiety or depression symptoms (671)	224 (33.4%)
Loneliness or social isolation (671)	177 (26.4%)
Cognition [GDS] (671)	
GDS 1	136 (20.3%)
GDS 2	60 (8.9%)
GDS 3	62 (9.2%)
GDS 4	87 (13.0%)
GDS 5	76 (11.3%)
GDS 6	161 (24.0%)
GDS 7	89 (13.3%)
BADL dependance [Barthel] (671)	36.3 (±26.9)
Bathing	579 (86.3%)
Climbing stairs	548 (81.7%)
Dressing	526 (78.4%)
Personal hygiene	486 (72.4%)
Toileting	434 (64.7%)
Walking	356 (53.1%)
Wheelchair handling (287)	237 (82.6%)
Transferring	344 (51.3%)
Eating	173 (25.8%)
**Institutionalization variables **	
Residents’ outdoor accessibility (671)	655 (97.6%)
Resident’s garden-like area accessibility (655)	471 (71.9%)
Food service arrangement (671)	
Own kitchen	532 (79.3%)
Catering service	139 (20.7%)
Ownership (for profit) (671)	538 (80.2%)
NH size [number of beds] (630)	52.98 (±29.80)
Small (≤50 beds)	266 (42.2%)
Middle (51–100 beds)	329 (52.2%)
Large (˃100 beds)	35 (5.6%)
Shared rooms per total rooms [%] (630)	69.18 (±27.79)
Shared toilet per total toilets [%] (630)	52.42 (±32.54)
Registered nurses’ ratio [hours/resident] (587)	1.34 (±1.04)
Physical therapists’ ratio [hours/resident] (587)	0.58 (±0.42)
Occupational therapists’ ratio [hours/resident] (587)	0.19 (±0.17)
Psychologists’ ratio [hours/resident] (587)	0.33 (±0.23)
Social workers’ ratio [hours/resident] (587)	0.43 (±0.21)
Caregivers’ ratio [hours/resident] (587)	11.82 (±4.78)
Nursing assistants’ ratio [hours/resident] (587)	1.35 (±5.63)
Leisure monitors’ ratio [hours/resident] (587)	0.05 (±0.13)
Geriatricians’ ratio [hours/resident] (587)	0.19 (±0.12)
Total professionals’ ratio [hours/resident] (587)	16.26 (±5.30)

**Table 2 healthcare-12-00810-t002:** Factors Related to limitations in Early-Loss BADL (Self-care) in institutionalized older adults in Catalonia, Spain.

	Dependent	Independent/Minimal Help	*p*	OR (95% CI)	*p*	OR (95% CI)
Variable	Frequency (%) or Mean (SD)		(Bivariate Analysis)		(Multivariate Analysis)	
Sex (female)	447 (89.6%)	52 (10.4%)	0.929	0.97 (0.57–1.74)	0.066	0.54 (0.28–1.04)
Age	86.1 (±8.6)	69 (±83.4)	0.041	17,484.50	0.375	1.01 (0.98–1.04)
Urinary continence	349 (96.1%)	14 (3.9%)	<0.001	5.61 (3.05–10.33)	<0.001	1.42 (1.17–1.72)
Cognition (GDS 2–6)	497 (93.8%)	33 (6.2%)	<0.001	5.42 (3.23–9.11)	<0.001	1.67 (1.41–1.99)
Multiple falls or serious fall	80 (96.4%)	3 (3.6%)	0.031	3.41 (1.05–11.11)	0.050	3.42 (1.00–11.64)
Fecal continence	234 (98.7%)	3 (1.3%)	<0.001	14.26 (4.43–45.89)		
Residents’ outdoor accessibility	589 (90.6%)	61 (9.4%)	<0.001	9.66 (3.50–26.64)		
Dysphagia	84 (98.8%)	1 (1.2%)	0.003	11.16 (1.53–81.43)		
Ulcer or chronic wound	56 (98.2%)	1 (1.8%)	0.026	7.052 (0.96–51.76)		
Confusion or behavior disorder	148 (94.3%)	9 (5.7%)	0.029	2.20 (1.07–4.55)		
NH type (private)	471 (88.4%)	62 (11.6%)	0.031	0.42 (0.19–0.95)		
Intellectual disability	33 (80.5%)	8 (19.5%)	0.047	0.446 (0.20–1.01)		
Neurological disease	199 (93.0%)	15 (7.0%)	0.051	1.80 (0.99–3.27)		
Weight loss or appetite loss	94 (94.9%)	5 (5.1%)	0.060	2.39 (0.94–6.10)		
Days of room isolation due to COVID-19	21.3 (±28.0)	15.1 (±24.7)	0.065	15,851.00		
Education level (illiterate)	89 (85.6%)	15 (14.4%)	0.081	0.58 (0.31–1.08)		
Marital status (married or in a relationship)	100 (94.3%)	6 (5.7%)	0.083	2.11 (0.89–5.02)		
Institutionalization months	37.0 (±41.8)	31.6 (±45.0)	0.089	17,964.00		
Food service arrangement (catering service)	128 (92.8%)	10 (7.2%)	0.178	1.61 (0.80–3.24)		
					R^2^ = 28.0%	

CI: confidence interval; OR: odds ratio; SD: standard deviation.

**Table 3 healthcare-12-00810-t003:** Factors Related to limitations in Mid-Loss BADLs (Mobility) in institutionalized older adults in Catalonia, Spain.

	Dependent	Independent	*p*	OR (95% CI)	*p*	OR (95% CI)
Variable	Frequency (%) or Mean (SD)		(Bivariate Analysis)		(Multivariate Analysis)	
Sex (female)	304 (60.4%)	199 (39.6%)	0.374	1.17 (0.82–1.67)	0.827	0.95 (0.63–1.45)
Age	86.6 (±8.7)	84.8 (±8.8)	0.003	46,841.50	0.970	1.00 (0.99–1.03)
Fecal incontinence (occasionally or always)	205 (85.8%)	34 (14.2%)	<0.001	7.40 (4.92–11.15)	<0.001	1.41 (1.22–1.63)
Ulcer or chronic wound	51 (89.5%)	6 (10.5%)	<0.001	6.47 (2.74–15.31)	0.003	4.01 (1.60–10.03)
Cognition (GDS 2–6)	340 (63.6%)	195 (36.4%)	<0.001	2.28 (1.55–3.33)	0.008	1.14 (1.03–1.25)
Urinary incontinence (occasionally or always)	275 (75.1%)	91 (24.9%)	<0.001	4.56 (3.27–6.35)	0.020	1.17 (1.02–1.33)
Physical restraint	90 (8.2%)	12 (11.8%)	<0.001	6.48 (3.46–12.11)		
Severe mental disorder	48 (44.9%)	59 (55.1%)	<0.001	0.50 (0.33–0.75)		
Dysphagia	68 (79.1%)	18 (20.9%)	<0.001	2.89 (1.68–4.98)		
Residents’ outdoor accessibility	396 (60.5%)	259 (39.5%)	<0.001	6.63 (1.87–23.48)		
Confusion or behavior disorder	111 (69.4%)	49 (30.6%)	0.004	1.75 (1.20–2.55)		
Institutionalization months	38.0 (±41.2)	34.5 (±43.9)	0.023	48,356.50		
Constipation	170 (64.6%)	93 (35.4%)	0.023	1.25 (1.05–1.99)		
Caregivers per residents (%)	33.5 (±13.2)	32.2 (±13.0)	0.042	37,477.50		
Days of room isolation due to COVID-19	22.0 (±27.9)	19.0 (±27.5)	0.047	53,036.00		
Weight loss or appetite loss	68 (68.0%)	32 (32.0%)	0.059	1.54 (0.98–2.42)		
Intellectual disability	19 (46.3%)	22 (53.7%)	0.077	0.57 (0.30–1.07)		
Vision impairment (0–10)	0.9 (±2.8)	0.6 (±2.0)	0.081	51,318.00		
Anxiety or depression symptoms	123 (54.9%)	101 (45.1%)	0.089	0.76 (0.55–1.04)		
Food service arrangement (catering service)	91 (65.5%)	48 (34.5%)	0.105	1.38 (0.93–2.04)		
Neurological disease	138 (63.6%)	79 (36.4%)	0.132	1.29 (0.93–1.80)		
Resident COVID-19 infection history	259 (60.9%)	166 (39.1%)	0.185	1.25 (0.90–1.75)		
Polypharmacy	340 (58.5%)	241 (41.5%)	0.234	0.75 (0.47–1.20)		
Education level (illiterate)	68 (64.8%)	37 (35.2%)	0.250	1.33 (0.86–2.06)		

CI, confidence interval; OR, odds ratio; SD, standard deviation.

**Table 4 healthcare-12-00810-t004:** Factors Related to dependence in Late-Loss BADL (Eating) in institutionalized older adults in Catalonia, Spain.

	Dependent	Independent/Minimal Help	*p*	OR (95% CI) or U Mann-Whitney	*p*	OR (95% CI)
Variable	Frequency (%) or Mean (SD)		(Bivariate Analysis)		(Multivariate Analysis)	
Sex (female)	151 (30.0%)	352 (70.0%)	0.046	1.52 (1.01–2.29)	0.117	1.52 (0.90–2.56)
Age	85.9 (±8.8)	85.9 (±8.7)	0.947	45,159.0	0.152	0.98 (0.96–1.01)
Fecal incontinence (occasionally and always)	133 (55.6%)	106 (44.4%)	<0.001	8.74 (5.96–12.81)	<0.001	1.56 (1.38–1.77)
Food service arrangement (catering service)	67 (48.2%)	72 (51.8%)	<0.001	3.16 (2.14–4.67)	<0.001	4.28 (2.60–7.05)
Neurological disease	79 (36.4%)	138 (63.6%)	<0.001	1.81 (1.28–2.57)	<0.001	2.11 (1.36–3.28)
Cognition (GDS 2–6)	159 (38.5%)	254 (61.5%)	<0.001	2.66 (1.60–4.42)	<0.001	1.31 (1.16–1.48)
Dysphagia	59 (58.1%)	36 (41.9%)	<0.001	4.489 (2.81–7.18)	0.009	2.16 (1.21–3.87)
Polypharmacy	150 (25.8%)	431 (74.2%)	<0.001	0.47 (0.30–0.74)	0.016	0.49 (0.27–0.88)
Weight loss or appetite loss	41 (41.0%)	59 (59.0%)	0.002	2.004 (1.29–3.11)	0.026	1.90 (1.08–3.35)
Physical restraint	49 (48.0%)	53 (52.0%)	<0.001	2.96 (1.91–4.57)		
Confusion or behavior disorder	64 (40.0%)	96 (60.0%)	<0.001	2.08 (1.43–3.02)		
Ulcer or chronic wound	33 (57.9%)	24 (42.1%)	<0.001	4.063 (2.33–7.09)		
Urinary incontinence (occasionally or always)	143 (39.1%)	223 (60.9%)	<0.001	3.878 (2.630–5.72)		
NH type (private)	132 (24.5%)	406 (75.5%)	<0.001	0.45 (0.30–0.66)		
Shared toilets per total toilets (%)	59.3 (±31.0)	49.8 (±32.8)	<0.001	335,500.0		
Caregivers per residents (%)	35.3 (±11.0)	31.9 (±13.8)	<0.001	27,046.5		
Doctors per residents (%)	0.5 (±0.3)	0.4 (±0.3)	0.005	29,910.0		
Anxiety or depression symptoms	49 (21.9%)	175 (78.1%)	0.012	0.62 (0.43–0.90)		
Renal disease	28 (19.7%)	114 (80.3%)	0.013	0.57 (0.36–0.89)		
Residents’ garden-like area accessibility	147 (31.2%)	324 (68.8%)	0.016	1.633 (1.09–2.44)		
Shortness of breath (0–10)	0.2 (±1.0)	0.5 (±1.7)	0.046	43,354.5		
Marital status (married or in a relationship)	38 (35.8%)	68 (64.2%)	0.050	1.55 (1.00–2.40)		
Residents’ outdoor accessibility	187 (28.5%)	468 (71.5%)	0.050	5.994 (0.79–45.70)		
Days of room isolation due to COVID-19	22.2 (±27.3)	20.2 (±27.9)	0.069	36,710.5		
Institutionalization months	38.2 (±40.7)	35.9 (±43.0)	0.111	42,492.5		

CI: confidence interval; OR: odds ratio; SD: standard deviation.

## Data Availability

The data presented in this study are available on request from the corresponding author.

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
