# Peer review of "Exploring Early, Middle, and Late Loss in Basic Activities of Daily Living among Nursing Home Residents: A Multicenter Observational Study"

_healthcare, 2024, doi:10.3390/healthcare12080810_

Round 1

Reviewer 1 Report

Comments and Suggestions for Authors

This is a very interesting topic however research methods and case collection methods are very vague, especially how the researchers collect the data. This is a cross-sectional study, and the consistency of the researchers' methods of collection data must be clearly stated. Tables are difficult to read.

Reviewer 2 Report

Comments and Suggestions for Authors

Thank you very much for having the opportunity to review this paper. This study presents an interesting topic and has a large sample size. The manuscript is clear and very well written, with its limitations well defined and results discussion well-crafted. However, some questions need to be resolved before considering this manuscript ready for publication:

1. In lines 158-162 it is suggested to replace:

“Limitation in self-care (early loss) was considered when any of the following BADLs had a score below 4 or 5: personal hygiene, dressing, or bathing. Limitations in mobility was indicated when any of the following BADLs had a rating below 4 or 5: walking or wheelchair handling, toileting and transferring. Limitation in eating was inferred with ratings below 4 or 5 in the eating BADL”. 

by 

“Limitation in self-care (early loss) was considered when any of the following BADLs had a score below 4: personal hygiene, dressing, or bathing. Limitations in mobility was indicated when any of the following BADLs had a rating below 4: walking or wheelchair handling, toileting and transferring. Limitation in eating was inferred with ratings below 4 in the eating BADL”. 

2. It is suggested that the authors divide Table 1 into three tables (one for demographic data, one for health data and the last one for institutionalization data) or that Table 1 be subdivided and subtitled in order to make it easier for the reader to find the variables.

3. Line 236 says “... and 28.0% on eating (late loss BADL).” However, according to Table 1, the percentage for “Eating” is 25.8%. Why is there a difference of 2.2%?

4. In order to summarize all the results already presented, it is proposed to transfer the Figure 2 to below of results of the third model.

Comments on the Quality of English Language

Minor editing of English language is required.

Reviewer 3 Report

Comments and Suggestions for Authors

The article analyses a very topical and complex internal issue. It is an example of a social problem in which an excess of variables hinders the transparency of the study. However, the authors have dealt with this limitation very well and the results do not lose their quality.  The verified findings could be of great importance in changing the approach to long-term care towards a more rehabilitative and activation rather than behavioural approach (see: lifts!).  The article fits in with current research trends, it is interesting, correctly structured, methodologically correct and practically relevant. An attempt to reduce the repetitive listing of factors would have impacted on the ease of communication, but this is purely an editorial comment.

Reviewer 4 Report

Comments and Suggestions for Authors

Congratulations on the presented text. It's a very current topic that appeals to a broad group of researchers with significant practical application.

Introdution

Line 45 - Global population is aging due to increased life expectancy and declining birth rates

I believe the sentence could be potentially confusing. Global population is aging due to increased life expectancy, not due to declining birth rates.
